# Semi-supervised multi-organ segmentation with cross supervision using siamese network

Jia Dengqiang[1],ORCID=0000-0002-0902-1882

1, School of Naval Architecture, Ocean and Civil Engineering, Shanghai Jiao Tong University, Shanghai, China, `wangxifeng004@163.com`

**Abstract.** Numerous unlabeled data is useful for supervised medical image segmentation, if the labeled data is limited. To leverage all the unlabeled images for efficient abdominal organ segmentation, we developed semi-supervised framework with cross supervision using siamese network, i.e., SemiSeg-CSSN. Cross supervision enables the two networks to optimize the network using pseudo-labels generated by the other. Moreover, we applied the cascade strategy for the task because of the large and uncertain locations of the abdomen regions. To validate the effects of unlabeled data, we employed an unlabeled image filtering strategy to select the unlabeled image and their pseudo label images with low uncertainty. On the FLARE2022 validation cases, with the help of unlabeled data, our method obtained the average dice similarity coefficient (DSC) of 77.7% and average normalized surface distance (NSD) of 82.0%, which is better than the supervised method. The average running time is 12.9s per case in inference phase and maximum used GPU memory is 2052 MB.

**Keywords:** Semi-supervised, cross supervision · label filtering

## 1 Introduction

For supervised learning, few labeled data tend to leading the over-fitting problem. In the task of medical image analysis, however, manual voxel-level labeling is expensive and time-consuming because of the professional domain knowledge.

Semi-supervised learning (SSL) aims to solve the learning problem in scenario of sparsely labeled images and a large number of auxiliary unlabeled images. These learning methods have been studied in classification problems [8,15].

Currently, semi-supervised segmentation has raised attention. Self-training strategy tries to learn from unlabeled data by imputing the labels for samples predicted with high confidence [1,2,14].

There are many datasets of natural images datasets available for semi-supervised segmentation, such as Pascal VOC 2012 [7] and Cityscapes [6]. For medical image segmentation, FLARE2022 challenge has a large-scale abdominal datasets that contains 50 labeled images and 2000 unlabeled images. Besides, the challenge has at most 13 organs are annotated, which belongs to the standard closed-set

SSL [4]. The difficulty of this challenge is to segment both large and small organs given a scenario with less labeled data.

The multi-organ segmentation have three main difficulties.

- Class imbalance problem.As shown in Figure 1, RAG and LAG have small class ratios, which leads to the class imbalance problem.
- Large shape variations and pathology influence. Some organs, e.g., Gallbladder (brown), Pancreas (yellow) and Duodenum (blue) shown in Figure 2, have large variation on shapes, and some organs are diseased, such as liver and kidney tumors (see Figure 3).
- Non-uniform images. Some images have incomplete abdominal regions, and the image information are not normalized.

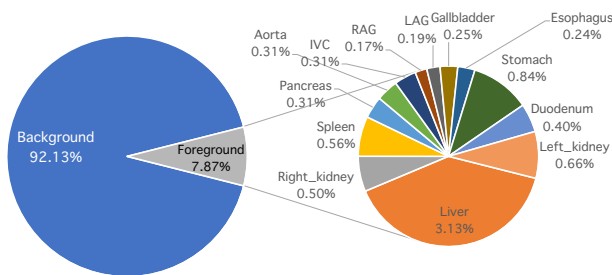

**Fig. 1.** Class ratio in the 50 labeled images in FLARE2022.

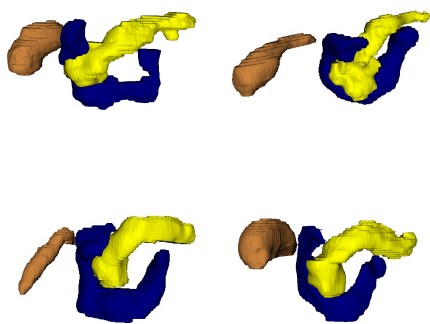

**Fig. 2.** Four selected examples in FLARE2022. Large shape variations of Gallbladder (brown), Pancreas (yellow) and Duodenum (blue).

In this paper, we proposed a siamese network with cross supervision to train the semi-supervised segmentation network. Two networks, which have the same

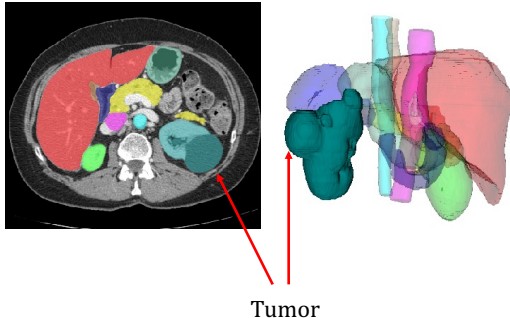

Tumor

**Fig. 3.** A selected examples in FLARE2022. The left kidney has a tumor.

architecture and the same number of parameters, are introduced, and they are initialized differently at the beginning of training. These two segmentation networks can generate pseudo label images, and supervise each other's training in the manner of cross supervision. Moreover, we employed a filtering strategy for unlabeled images. These selected unlabeled images have pseudo label images with low uncertainty, which can ensure the stability of training.

The main contributions of our work are summarized as follows:

- To leverage the unlabeled data, we use cross supervision strategy, which is achieved via a siamese network.
- To improve the efficiency, we use anisotropic convolution block and strip pooling module.
- We also employ a filtering strategy to improve the performance.
- The effectiveness and efficiency of the proposed semi-supervised framework are demonstrated on FLARE2022 challenge dataset, where we achieve the top 10 with low time cost and less memory usage.

## 2   Method

Figures 4 and 5 show our approach using cross supervision with labelled and unlabelled data, respectively.

### 2.1   Preprocessing

The labeled images are cropped using their corresponding labels, which avoid selecting the patch without any labels. For the unlabeled images, we use the trained coarse segmentation model to crop the abdominal regions. All the images are re-sampled for a fixed spacing, i.e., 1mm × 1mm × 3mm.

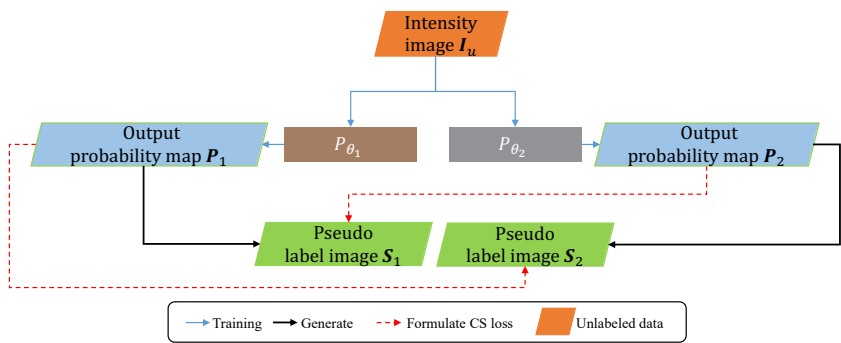

**Fig. 4.** Cross supervision framework when using unlabeled data. The siamese network contains two sub-networks, denoted as $P_{\theta_1}$ and $P_{\theta_2}$, whose architectures are the same, and the two sub-networks are initialized differently at the beginning of training. When using unlabeled images, two segmentation probability maps $P_1$ and $P_2$ for the given intensity image $\boldsymbol{I}_u$. The two probability maps can be transformed to two different pseudo label images $S_1$ and $S_1$ for the input image. Thus, these two segmentation sub-networks can generate pseudo label images, and supervise each other's training in the manner of cross supervision.

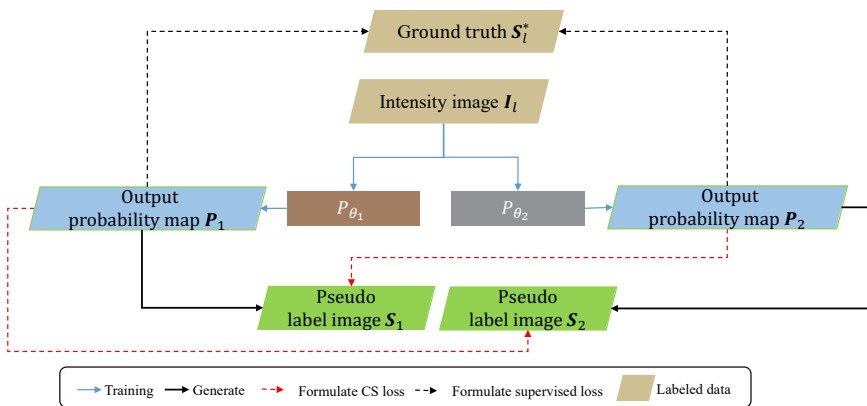

**Fig. 5.** Cross supervision framework when using labeled data. When using labeled images, two loss functions are constructed, i.e., the cross supervision loss and supervision loss. The two subnetworks can also generate the pseudo label images for intensity image $\boldsymbol{I}_l$. The two subnetworks can be supervised based on these pseudo label images in the cross-supervision manner. Besides, the label images can be considered as ground truth of segmentations, therefore, the output probability maps can be also supervised with the ground truth of the segmentation.

## 2.2 Proposed Method

We propose a semi-supervised segmentation framework for multi-organ segmentation task, which can leverage large number of unlabeled data. The framework consists of two sub-networks, which have the same structures. We separately optimize the sub-networks, and simultaneously use them to predict the pseudo labels of unlabeled data. We train the two networks in a cross-supervised manner [3].

Besides, we employed a cascade strategy, which aims to segmenting the abdomen organs via coarse-to-fine procedure [18]. Because the region of interest, i.e., ROI, of abdominal organs is large, we can not efficiently segment all the organs in a single-stage network. Therefore, we first segmented the organs from downsampling images, which can be seen as a coarse segmentation. With the help of the coarse segmentation, we segmented the organs from the original images in the second stage.

We can also train a semi-supervised network via selected unlabeled images, which is based on the uncertainty metric of their pseudo label images.

## 2.3 Cross supervision using siamese network (CSSN)

Let $\mathcal{L} = \{(I_1, S_1^*), (I_2, S_2^*), ..., (I_N, S_N^*)\}$ and $\mathcal{U} = \{I_{N+1}, I_{N+2}, , ..., I_M\}$ denote the labeled data and unlabeled data. $I$ and $S^*$ denote the intensity image and label image. The aim of the semi-supervised segmentation is to obtain a segmentation plan that can leverage $\mathcal{L}$ and $\mathcal{U}$. We can use the segmentation plan to predict a probability map $P$ for $I$ as:

$$P = P_\theta(I). \tag{1}$$

In particular, we introduce two sub-networks, i.e., $P_{\theta_1}$ and $P_{\theta_2}$, to obtain two probability maps for a fixed image $I$:

$$\begin{aligned} P_1 &= P_{\theta_1}(I), \\ P_2 &= P_{\theta_2}(I). \end{aligned} \tag{2}$$

The siamese networks ($P_{\theta_1}$ and $P_{\theta_1}$) have the same structures and the same number of parameters but are initialized differently at the beginning of training. As in Equation 2, we can obtain two different predictions for one input image because of the two sub-networks with different parameters. Since we use label information for supervision, we can use any supervised loss functions, e.g., cross-entropy loss, dice loss and combination of them, which we denote as $\ell s$ in this work.

As shown in Figure 4, for unlabeled data $\mathcal{L}$, we employ a bidirectional consistent strategy for supervision. For example, sub-network $P_{\theta_1}$ can be supervised by the pseudo label images generated by the frozen subnetwork $P_{\theta_2}$. For voxel $i$ of an unlabeled image, we can calculate the loss as:

$$\ell_s\left(p_{i|\theta_1}, s_{i|\theta_2}\right), \tag{3}$$

where $p_{i|\theta_1}$ denotes the predicted probability for voxel $i$ using sub-network $P_{\theta_1}$, and $s_{i|\theta_2}$ is the label for voxel $i$ generated by $P_{\theta_2}$. The sub-network $P_{\theta_2}$ can be supervised in the similar manner.

Thus, the cross-supervised (CS) loss function for $\boldsymbol{I}_u \in \mathcal{U}$ can be formulated as follows:

$$\mathcal{C}_u^u = \frac{1}{V_u} \sum_{i \in \boldsymbol{I}_u} \left( \ell_s \left( p_{i|\theta_1}, s_{i|\theta_2} \right) + \ell_s \left( p_{i|\theta_2}, s_{i|\theta_1} \right) \right), \tag{4}$$

where $V_u$ is the number of voxels in $\boldsymbol{I}_u$.

As shown in Figure 5, for labeled data $\boldsymbol{I}_l \in \mathcal{L}$, we first employ supervised loss functions for the two sub-networks:

$$\mathcal{C}_s = \frac{1}{V_l} \sum_{i \in \boldsymbol{I}_l} \left( \ell_s \left( p_{i|\theta_1}, s_i^* \right) + \ell_s \left( p_{i|\theta_2}, s_i^* \right) \right), \tag{5}$$

where $s_i^*$ is the voxel $i$ in the label image $\boldsymbol{S}^*$. The number of voxels in $\boldsymbol{I}_l$ is denoted as $V_l$.

As shown in Figure 5, using the pseudo label images, we can also formulate the CS loss for labeled data in the same manner as Equation (4), i.e., $\mathcal{C}_u^l$.

The training loss function can be formulated as:

$$\mathcal{C} = \mathcal{C}_s + \mathcal{C}_u^u + \mathcal{C}_u^l. \tag{6}$$

### 2.4   Unlabeled image filtering (UIF) based on uncertainty

Unlabeled images may contain cases with different distributions than labeled images. Although we used pseudo-label images inject strong data augmentations, some pseudo-label images with high uncertainty were still prone to accumulate and degrade the performance. To solve this problem, we prioritized reliable unlabeled images based on holistic prediction-level stability.

To obtain reliable unlabeled images, we employed UIF on the pseudo-labeled images, as the selection approach of Yang et al. [17]. We selected 200 (top 10%) unlabeled images and their pseudo-labeled images with the lowest uncertainty from the 2000 unlabeled images. For an unlabeled image $\boldsymbol{I}_l$, we can compute the uncertainty as:

$$U_l = 1 - \frac{1}{9} \sum_{j=1}^{9} \text{DSC}(\boldsymbol{S}_{l|\mathcal{M}_{j*100}}, \boldsymbol{S}_{l|\mathcal{M}_{1000}}), \tag{7}$$

where $\boldsymbol{S}_{l|\mathcal{M}_{j*100}}$ denotes the pseudo label image of $\boldsymbol{I}_l$ generated by trained model $\mathcal{M}_{j*100}$. The trained model $\mathcal{M}_{j*100}$ is saved to the disk in epoch $j*100$ during training the supervised segmentation network.

After we have selected 200 unlabeled images, we can train a new SemiSeg-CSSN with a new mixed dataset containing both labeled and unlabeled images. We embedded CSSN during training the network.

### 2.5   Strategies to improve inference speed and reduce resource consumption

We take the whole image as input and output a segmentation result of the whole image size, which is more efficient than using a patchwork segmentation result based on patches. Besides, we employed the strategies from efficientSegNet [18] to reduce the resource consumption. An anisotropic convolution with a $k \times k \times 1$ intra-slice convolution and a $1 \times 1 \times k$ inter-slice convolution are used in the decoder module. In addition, the low-level and high-level feature maps are aggregated by addition rather than concatenation due to the low GPU memory footprint.

### 2.6   Post-processing

For the results of segmentations, we used the maximal union region selection as post-processing steps. We selected the unique region which has the maximal areas from the candidate regions for each class.

## 3   Experiments

### 3.1   Dataset and evaluation measures

The FLARE2022 dataset is curated from more than 20 medical groups under the license permission, including MSD [16], KiTS [9,10], AbdomenCT-1K [13], and TCIA [5]. It is an extension of the FLARE 2021 [12] with more segmentation targets and more diverse abdomen CT scans. The training set includes 50 labelled CT scans with pancreas disease and 2000 unlabelled CT scans with liver, kidney, spleen, or pancreas diseases. The validation set includes 50 CT scans with liver, kidney, spleen, or pancreas diseases. The testing set includes 200 CT scans where 100 cases has liver, kidney, spleen, or pancreas diseases and the other 100 cases has uterine corpus endometrial, urothelial bladder, stomach, sarcomas, or ovarian diseases. All the CT scans only have image information and the center information is not available.

The evaluation measures consist of two accuracy measures: Dice Similarity Coefficient (DSC) and Normalized Surface Dice (NSD), and three running efficiency measures: running time, area under GPU memory-time curve, and area under CPU utilization-time curve. All measures will be used to compute the ranking. Moreover, the GPU memory consumption has a 2 GB tolerance.

### 3.2   Implementation details

**Environment settings** The environments and requirements for training are presented in Table 1.

**Table 1.** Environments and requirements for training.

| | |
|---|---|
| Windows/Ubuntu version | Ubuntu 20.04.4 LTS |
| CPU | Platinum 82 series (72vCPU) v5@2.5GHz |
| RAM | 16×4GB; 2.67MT/s |
| GPU (number and type) | NVIDIA V100 16×32G |
| CUDA version | 11.1 |
| Programming language | Python 3.6 |
| Deep learning framework | Pytorch (Torch 1.8.0, torchvision 0.9.0) |
| Code is publicly available at SemiSeg-CSSN | |

**Training protocols** We implemented the proposed framework using EfficientSeg-Net network used in FLARE21 challenge. The patch-based Unet such as nnUnet [11] also can be used as the basic segmentation, however, we found it consumes large RAM when prediction. Brightness, crop, random rotation, random transition and random elastic deformation were used for data augmentation. We random resampled the data with size described in Table 2. Besides, we trained the coarse model with the 50 labeled images.

**Table 2.** Training protocols for SemiSeg-CSSN.

| | |
|---|---|
| Network initialization | Kaiming normal initialization |
| Batch size | 8(coarse), 1(fine) |
| Input size (coarse) | 160×160×160 |
| Input size (fine) | 192×192×192 |
| Total epochs | 500(coarse), 200(fine) |
| Optimizer | Adam with betas (0.9, 0.99), L2 penalty: 0.00001 |
| Loss | Dice loss and focal loss (alpha = 0.5, gamma = 2) |
| Initial learning rate (lr) | 0.01 |
| Training time (coarse) | 6 (coarse), 300(fine) hours |

## 4    Results and discussion

### 4.1    Quantitative results on validation set

We used 50 labeled and 2000 unlabeled images to train the network in cross-supervised manner. The results show that the method using unlabeled data improve the dice score of the method with only 50 labeled images.

Table 3 shows the results of the proposed methods. The results of our submitted solution (docker container), which is evaluated by the organizers of FLARE2022,

**Table 3.** Quantitative results of supervised and semi-supervised methods in terms of DSC and NSD on the validation dataset. The symbol 50(L)+ 2000(U) denotes the method, which used 50 labeled and 2000 unlabeled images. We reported the mean and standard deviation in parentheses.

| Organ | Supervised 50(L) DSC(%), NSD(%) | SemiSeg-CSSN 50(L)+2000(U) DSC(%), NSD(%) | SemiSeg-CSSN+UIF 50(L)+200(U) DSC(%), NSD(%) | SemiSeg − CSSN* 50(L)+2000(U) DSC(%), NSD(%) |
|---|---|---|---|---|
| Liver | 90.8(7.3),85.4(13.0) | 93.7(5.4), 91.3(10.7) | 96.3(2.0),96.7(4.7) | 92.5(14.1),90.6(15.4) |
| RK | 79.3(33.6) ,77.93(32.8) | 80.5(32.8),79.0(32.5) | 79.8(31.9), 79.5(31.9) | 87.3(26.2),86.7(26.3) |
| Spleen | 92.5(7.4),90.0(12.3) | 89.8(21.9),89.0(23.4) | 87.6(18.7), 88.6(18.5) | 91.6(19.8),91.7(20.9) |
| Pancreas | 68.8(12.3),76.2(14.5) | 73.3(13.8),81.1(14.6) | 78.4(15.3), 88.8(13.8) | 75.4(15.8),82.8(16.9) |
| Aorta | 90.6(4.0),91.8(6.6) | 94.5(2.6),96.3(3.9) | 93.8(2.6), 96.5(3.2) | 91.8(14.5),93.6(15.2) |
| IVC | 83.5(11.7),80.4(12.3) | 87.7(8.6),87.0(8.9) | 87.0(11.6),87.1(10.6) | 82.4(16.2),80.5(17.0) |
| RAG | 54.1(31.6),63.0(35.0) | 64.1(33.2),72.6(37.1) | 77.3(14.6),89.9(10.0) | 64.8(31.7),74.5(36.0) |
| LAG | 24.4(29.6),48.1(34.8) | 51.7(32.0),71.6(27.8) | 72.1(20.7),82.4(21.7) | 47.5(32.9),70.0(30.8) |
| Gallbladder | 42.3(40.0),39.3(39.3) | 60.3(38.0),57.6(38.1) | 63.9(41.4),61.9(42.2) | 68.0(34.8),66.1(35.7) |
| Esophagus | 75.1(17.1),84.8(18.2) | 82.4(9.8),91.7(9.5) | 76.7(12.9),86.5(14.2) | 76.0(21.3),84.1(23.2) |
| Stomach | 71.0(29.0),71.0(29.3) | 83.0(18.8),84.8(17.9) | 74.2(28.7),78.5(28.1) | 80.8(24.2),82.4(24.4) |
| Duodenum | 61.7(26.0),76.9(26.0) | 64.0(23.8),78.2(22.0) | 59.1(26.8),74.8(24.9) | 63.3(23.1),77.0(22.7) |
| LK | 88.2(22.2),87.2(22.5) | 88.6(21.1),85.4(22.2) | 81.7(25.1), 82.3(25.0) | 88.6(19.9),85.4(21.5) |
| Avg. | 70.9(31.0) ,74.8(29.4) | 78.0(26.6),82.0(25.3) | 79.1(24.5),84.1(23.7) | 77.7(13.1),82.0(7.9) |

**Table 4.** Quantitative .

| Mean runtime (s) | Maximum used GPU memory (MB) | AUC GPU time | AUC CPU time |
|---|---|---|---|
| 12.9 | 2052 | 13776.9 | 250.6 |

are reported in the last two columns in Table 3, i.e., SemiSeg − CSSN*. The other results are evaluated on the 20 selected validation cases, whose ground truth are send by the organizers.

Compared to the supervised method, the average DSC of the semi-supervised method (SemiSeg-CSSN) improves from 70.9% to 78.0%, while the average NSC improves from 74.8% to 82.0%. The results show that LAG, Gallbladder and RAG segmentation is the three difficult organs and Liver, Spleen and Aorta is the three easy organs for abdominal organ segmentation. The difficulties may be due to unclear boundaries and class imbalance issues. Besides, the standard deviations of Gallbladder segmentation are relative large, which demonstrates the method achieves disappointed robustness for Gallbladder. As shown in Figure 6, Case #0047 has a complete Gallbladder, while Case #0048 does not have one. Moreover, as shown in Figure 6, the pathologies, such as the tumor in Liver in Case #0047, have negative effects on the segmentation.

Besides, for 2000 unlabeled images, we generated their pseudo label images using trained supervised segmentation network. Then, we used UIF to select 200 unlabeled images and their pseudo label images with low uncertainty, and trained a new segmentation network via SemiSeg-CSSN. As shown in Table 3, the DSC of semi-supervised segmentation network improves from 78.0% to 79.1%.

**Table 5.** Quantitative results SemiSeg-CSSN in terms of DSC and NSD on the test dataset. We used 50 labeled and 2000 unlabeled images for training. We reported the mean and standard deviation in parentheses.

| Organ | DSC(%) | NSD(%) |
|---|---|---|
| Liver | 94.3(5.2) | 92.7(8.8) |
| RK | 89.1 (21.6) | 87.6 (22.3) |
| Spllen | 90.6(20.3) | 90.7(21.3) |
| Pancreas | 71.2(17.9) | 79.8(19.3) |
| Aorta | 93.3(8.2) | 95.1(9.0) |
| IVC | 83.6(15.1) | 83.1(16.3) |
| RAG | 75.2(20.4) | 86.2(22.1) |
| LAG | 48.7(31.9) | 74.8(28.7) |
| Gallbladder | 66.5(36.3) | 65.0(36.2) |
| Esophagus | 69.0(22.4) | 77.5(25.3) |
| Stomach | 83.4(17.8) | 84.8(17.7) |
| Duodenum | 65.1(17.3) | 79.0(18.0) |
| LK. | 86.4(21.7) | 82.3(23.0) |
| Avg. | 78.2(13.0) | 83.0(7.7) |

**Table 6.** Quantitative results of the good (Case #0006 and Case #0035) and bad (Case #0047 and Case #0048) examples.

| Organ | Case #0006 DSC(%), NSD(%) | Case #0035 DSC(%), NSD(%) | Case #0047 DSC(%), NSD(%) | Case #0048 DSC(%), NSD(%) |
|---|---|---|---|---|
| Liver | 96.8, 93.2 | 96.7, 96.8 | 82.2, 67.0 | 87.1 , 82.2 |
| RK | 96.5, 96.7 | 97.3, 97.1 | 95.7, 93.4 | 84.8, 77.6 |
| Spleen | 97.9, 97.9 | 98.0 , 99.6 | 85.8, 76.4 | 63.6, 50.0 |
| Pancreas | 83.6, 90.5 | 87.9, 99.0 | 60.9 , 68.0 | 66.6, 75.7 |
| Aorta | 95.8, 97.6 | 96.4, 99.8 | 92.3, 96.8 | 86.9, 85.2 |
| IVC | 94.5, 97.5 | 93.5 , 94.4 | 86.0, 84.7 | 53.5, 57.7 |
| RAG | 90.9, 97.6 | 70.2, 85.8 | 0.0,0.0 | 63.7, 66.0 |
| LAG | 87.1, 95.2 | 75.3, 80.6 | 11.5, 61.5 | 15.0, 63.9 |
| Gallbladder | 100.0, 100.0 | 53.9, 52.9 | 52.7, 55.0 | 0.0, 0.0 |
| Esophagus | 85.4, 92.9 | 88.52, 96.6 | 87.2 , 99.0 | 57.6, 70.8 |
| Stomach | 89.9, 88.6 | 92.8, 97.7 | 76.4, 76.8 | 40.3, 41.0 |
| Duodenum | 72.9, 82.67 | 85.0, 96.8 | 4.8, 15.3 | 59.5, 76.4 |
| LK | 95.1, 88.2 | 97.6, 98.3 | 98.1 , 98.2 | 93.8, 89.4 |
| Avg. | 91.3, 93.6 | 87.2, 91.6 | 64.1, 68.3 | 59.4, 63.9 |

### 4.2   Segmentation efficiency results on validation set

Table 4 presents the segmentation efficiency results. The mean runtime is 12.9 s per case in prediction step, maximum used GPU memory is 2052 MB, AUC GPU time is 13776.9, and AUC CPU time 250.6.

### 4.3   Quantitative results on test set

Table 5 shows the quantitative result of SemiSeg-CSSN on test dataset. The average DSC of 13-organ segmentation is $78.2 \pm 13.0\%$, and the average NSD is $83.0 \pm 7.8\%$. The organs with the highest and lowest DSC were Liver and LAG, respectively. The gallbladder has the largest standard deviation.

### 4.4   Ablation study: influence of different number of unlabeled data

To further validate the effect unlabeled images, different numbers of unlabeled images from the training set were selected. For each case, we trained the

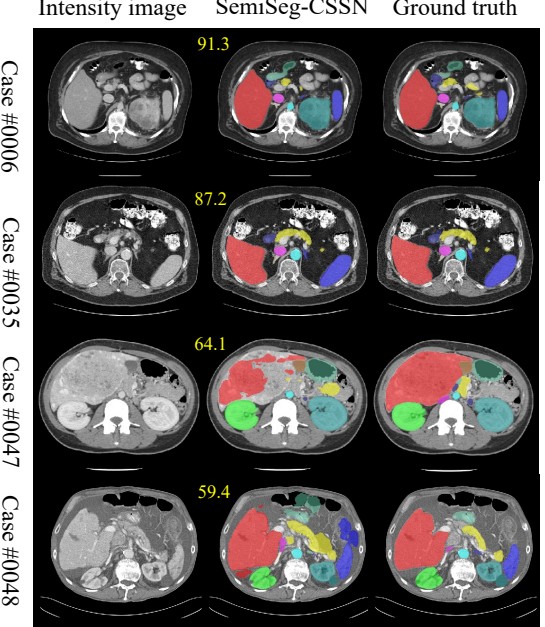

**Fig. 6.** Qualitative results on good (Case #0006 and Case #0035) and bad (Case #0047 and Case #0048) examples. First column is the image, second column is the results achieved by our propose method, and third column is the ground truth of the segmentation. The DSC of each case is presented at the top-left corner.

SemiSeg-CSSN model using 50 labeled images and different number, i.e., ranging from 0 to 2000, of unlabeled images. Note that supervised model used 0 unlabeled images. Figure 7 shows the segmentation results. With the number of unlabeled images increases, the performance of SemiSeg-CSSN models are increased, and all the models with unlabeled images perform better than the supervised method. Moreover, it is clear that the SemiSeg-CSSN model tends to converge when trained with more than 1000 unlabeled images.

## 5    Discussion and conclusion

Using unlabeled data, the proposed semi-supervised method achieved better results than the results of the supervised method. Whether using supervised or semi supervised methods, the segmentation of some organs is still challenging. LAG segmentation obtained disappointing performance because of unclear boundaries and class imbalance issues. The existence of seriously pathology-affected organs, such as Livers and Kidneys, are critical factor for the poor segmentation performance. Besides, further research is needed to identify accurate boundaries and suppress pathological effects.

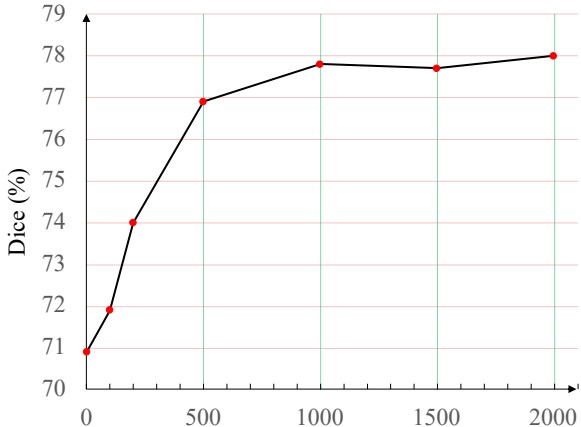

**Fig. 7.** Performance plot of our semi-supervised approach with 50 labeled images and different number (from 0 to 2000) of unlabeled images. Note the case with 0 unlabeled images denotes the fully supervised method.

SemiSeg-CSSN+UIF model only used 200 unlabeled images, which achieved higher DSC and NSD than the model with un-filtering 2000 unlabeled images. It means the quality counts more than quantity when using unlabeled images. However, because UIF can be used in any trained segmentation network to select unlabeled images, there is a progressive training strategy, which is needed to explore in the future.

### 5.1   Limitation and future work

We summarize the limitations and potential improvement as follows:

– Address the difficulties of multi-organ segmentation with class imbalance problem.
– Robust algorithms for shape variation of organs and presence of pathologies.
– Normalization of ROI and image information.
– The quality of pseudo labels needs more attention than quantity.

**Acknowledgements** The authors of this paper declare that the segmentation method they implemented for participation in the FLARE 2022 challenge has not used any pre-trained models nor additional datasets other than those provided by the organizers.

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
