# OpenReview forum: "Semi-supervised multi-organ segmentation with cross supervision using siamese network"
_MICCAI.org/2022/Challenge/FLARE_

### Official Review · Reviewer_Yu3W · 2022-09-13
**In this paper, the authors proposed a siamese network with cross supervision to train the semi-supervised segmentation network. This paper is technically well written, further supported by preprocessing, postprocessing, and ablation study, and can be accepted for publication; however, there are some grammatical mistakes and typos.**

**Rating:** 7
**Confidence:** 3

**Review:**

In this paper, the authors proposed a siamese network with cross supervision to train the semi-supervised segmentation network. They introduced two same architectures, which were initialized differently. Both networks can generate pseudo label images and supervise each other’s training through cross supervision.

This paper is technically well written, further supported by preprocessing, postprocessing, and ablation study, and can be accepted for publication; however, there are some grammatical mistakes and typos, such as

(I)  Nonuniform should be non-uniform

(II) Two networks with the same architecture are introduced, and the are initialized differently at the beginning of training (rewrite this sentence)

(III) subnetworks should be sub-networks

(IV) We train the two networks in a cross-supervised way (we should write like “We train the two networks in a cross-supervised manner”)

(V) Please check the whole manuscript for such types of problems and modify it accordingly.

(VI) Fig-2 entails three sub-images; therefore, it is suggested to assign valid numbers to each sub-image, such as (a), (b), (c) or (i), (ii), and (iii) with a clear caption. Likewise, Fig-3 and 4 need some proper explanation.

---

> ### Author Response · Authors · 2022-10-20
> **We have revised manuscript and provide a point-by-point response to each of the issues raised by reviewer Yu3W (Reviwer #1)**
>
> Comment1.1: In this paper, the authors proposed a siamese network with cross supervision to train the semi-supervised segmentation network. They introduced two same architectures, which were initialized differently. Both networks can generate pseudo label images and supervise each other’s training through cross supervision.
> This paper is technically well written, further supported by preprocessing, postprocessing, and ablation study, and can be accepted for publication; however, there are some grammatical mistakes and typos,
>
> Response1.1: We thank the reviewer for these insightful and important comments. Please see the detailed responses below.
>
> Comment1.2: Nonuniform should be non-uniform
>
> Response1.2: Thanks for the comments. We have modified the words in the revised manuscript.
>
> Comment1.3: Two networks with the same architecture are introduced, and the are initialized differently at the beginning of training (rewrite this sentence)
>
> Response1.3: Thanks for the comments. We have rewrite this sentence in the revised manuscript.
>
> Comment1.4: Subnetworks should be sub-networks
>
> Response1.4: Thanks for the comments. We have modified the word subnetwork as sub-network in the revised manuscript.
>
> Comment1.5: We train the two networks in a cross-supervised way (we should write like “We train the two networks in a cross-supervised manner”)
>
> Response1.5: Thanks for the comments. We have modified the words in the revised manuscript.
>
>
> Comment1.6: Please check the whole manuscript for such types of problems and modify it accordingly.
>
> Response1.6: Thanks for the comments. We have modified the words and sentences in the revised manuscript.
>
> Comment1.7: Fig-2 entails three sub-images; therefore, it is suggested to assign valid numbers to each sub-image, such as (a), (b), (c) or (i), (ii), and (iii) with a clear caption. Likewise, Fig-3 and 4 need some proper explanation.
>
> Response1.7: Thanks for the comments. We have divided Figure 2 into two figures in the revised manuscript. Moreover, we have added more explanations for Figure 3 and Figure 4.

---

### Official Review · Reviewer_cboQ · 2022-09-14
**There are some language and formatting issues waiting to be fixed**

**Rating:** 6
**Confidence:** 3

**Review:**

This paper studied the Semi-supervised problem in multi-organ segmentation. The author used Cross supervision method to achieve good results in the FLARE2022 challenge.

In the method section, the author's description is explained in a clear and readable way. Furthermore, this paper is well structured and the experiments are sufficient.

There are some problems, which must be solved before it is considered for publication. If the following problems are well-addressed:

* It is noted that your manuscript needs careful editing by someone with expertise in technical English editing paying particular attention to English grammar, spelling, and sentence structure so that the goals and results of the study are clear to the reader.

* The caption section of the figure needs to be supplemented with a more detailed description.

* Please recheck the paper with the recommendations in the official supplemental documentation.

---

> ### Author Response · Authors · 2022-10-20
> **We have revised manuscript and provide a point-by-point response to each of the issues raised by reviewer cboQ (Reviwer #2)**
>
> Comment2.1: This paper studied the Semi-supervised problem in multi-organ segmentation. The author used Cross supervision method to achieve good results in the FLARE2022 challenge.
> In the method section, the author's description is explained in a clear and readable way. Furthermore, this paper is well structured and the experiments are sufficient.
> There are some problems, which must be solved before it is considered for publication. If the following problems are well-addressed:
>
> Response2.1: We thank the reviewer for the important comments. Please see the detailed responses below.
>
> Comment2.2: It is noted that your manuscript needs careful editing by someone with expertise in technical English editing paying particular attention to English grammar, spelling, and sentence structure so that the goals and results of the study are clear to the reader.
>
> Response2.2: Thanks for the comments. We have modified the words and sentences in the revised manuscript.
>
> Comment2.3: The caption section of the figure needs to be supplemented with a more detailed description.
>
> Response2.3: Thanks for the comments. We have divided Figure 2 into two figures in the revised manuscript. Moreover, we have added more explanations for Figure2, Figure 3 and Figure 4 in the revised manuscript.
>
> Comment2.4: Please recheck the paper with the recommendations in the official supplemental documentation.
>
> Response2.4: Thanks for the comments. We have added the section of segmentation efficiency results and limitation and future work in the revised manuscript. Moreover, we have modified the words and sentences in the revised manuscript.

---

### Official Review · Reviewer_EtEE · 2022-09-15
**Great work but leave something to be describe and refine**

**Rating:** 7
**Confidence:** 3

**Review:**

1. In 2.3, Author said: The siamese networks (Pθ1 and Pθ1 ) have the same structures and parameters but are initialized differently at the beginning of training. I think the sentence should be the networks have the same structures and same number of parameters is better for understanding.

2. In 2.3, Author didn't explain what is $ V_u $.

3. Fig 5, In the figure, the order of coloumn is Intensity Image, SemiSeg-CSSN, Ground truth, but in description the author said the second column is ground truth is kinda confuse.

Over all, This paper is quite clear about author's work but still need refine.

---

> ### Author Response · Authors · 2022-10-20
> **We have revised manuscript and provide a point-by-point response to each of the issues raised by reviewer EtEE (Reviwer #3)**
>
> Comment3.1: In 2.3, Author said: The siamese networks (Pθ1 and Pθ1 ) have the same structures and parameters but are initialized differently at the beginning of training. I think the sentence should be the networks have the same structures and same number of parameters is better for understanding.
>
> Response3.1: Thanks for the comments. We have rewrite this sentence in the revised manuscript.
>
> Comment3.2: In 2.3, Author didn't explain what is V_u.
>
> Response3.2: Thanks for the comments. We have added the definition of symbols V_u and V_l in the revised manuscript.
>
> Comment3.3: Fig 5, In the figure, the order of coloumn is Intensity Image, SemiSeg-CSSN, Ground truth, but in description the author said the second column is ground truth is kinda confuse.
>
> Response3.3: We apologize the mistake. We have modified Figure 5 in the revised manuscript.
>
> Comment3.4: Over all, This paper is quite clear about author's work but still need refine.
>
> Response3.4: Thanks for the comments. We have modified the words and sentences in the revised manuscript.

---

### Official Review · Reviewer_xZU3 · 2022-09-15

**Rating:** 8
**Confidence:** 5

**Review:**

Nice overview of class imbalance in the introduction. The cross supervision setup is also interesting.

Some comments/suggestions

- Why did you only select 10% of the 2000 cases after uncertainty estimation? It seems like very few.

- Figure 6 is really nice

- Nice results overall, but I think your baseline performance could be better, for instance using nnU-Net

---

> ### Author Response · Authors · 2022-10-20
> **We have revised manuscript and provide a point-by-point response to each of the issues raised by reviewer xZU3 (Reviwer #4)**
>
> Comment4.1: Nice overview of class imbalance in the introduction. The cross supervision setup is also interesting.
> Some comments/suggestions
>
> Response4.1: We thank the reviewer for the important comments. Please see the detailed responses below.
>
> Comment4.2: Why did you only select 10% of the 2000 cases after uncertainty estimation? It seems like very few.
>
> Response4.2: Thanks for the comments. We select the cases after uncertainty estimation due to the idea that we want validate the quality of the pseudo label images needs more attention than the quantity of the pseudo label images. The proportion 10% is small, which is suitable to validate our idea.
>
> Comment4.3: Figure 6 is really nice
>
> Response4.3: Thanks for the comments.
>
> Comment4.4: Nice results overall, but I think your baseline performance could be better, for instance using nnU-Net
>
> Response4.4: Thanks for the comments. We agree that the baseline performance could be better. In our paper, we resampled all the data into a fixed spacing, i.e., 1×1×3mm. We use cropping strategy to focus on the abdominal region. Moreover, we use the efficientSegNet as our basic segmentation network, which resize the image into a fixed size, and input the whole image to the network for training. Based on this, we adapted dice loss and focal loss as the supervised loss function.
>     nnUnet is well known for its excellent performance and high robustness for various segmentation tasks. However, it is a patch-based segmentation network, which may take long time for prediction. Since FLARE2022 is a fast and low-resource semi-supervised challenge, the researchers should not only focus on segmentation accuracy but also on segmentation efficiency and resource consumption. If we use nnUnet as the basic segmentation network, we should modified the network architectures and reduce the parameters for fast predictions, which may cause accuracy decrease.

---

### Official Review · Reviewer_ESWe · 2022-09-16
**Well-written paper with solid experiment results**

**Rating:** 7
**Confidence:** 4

**Review:**

**Summary:**

This work proposed SemiSeg-CSSN for semi-supervised multi-organ segmentation and employed UIF to select high-quality unlabeled images. The method achieves an mDSC of 77.7% with low GPU consumption at 12.9s per image.

**Strengths:**

- Complete analysis of FRARE2022 competition train dataset.
- Employ coarse-to-fine network and anisotropic convolution to reduce resource consumption.
- Propose Cross supervision framework and unlabeled image filtering to obtain reliable unlabeled images for semi-supervised training.
- Ablation study for the number of unlabeled data: this experiment shows that UIF effectively selects high-quality unlabeled images.

**Suggest improvements:**

- Add description and figure for network architecture.
- The mDSC of baseline(supervised training) is about 80% or more in this competition, while it's 70.9% in this work. It is recommended to improve the performance of the baseline, and the result will be more credible.

---

> ### Author Response · Authors · 2022-10-20
> **We have revised manuscript and provide a point-by-point response to each of the issues raised by reviewer ESWe (Reviwer #5)**
>
> Comment5.1: This work proposed SemiSeg-CSSN for semi-supervised multi-organ segmentation and employed UIF to select high-quality unlabeled images. The method achieves an mDSC of 77.7% with low GPU consumption at 12.9s per image.
> Complete analysis of FRARE2022 competition train dataset.
> Employ coarse-to-fine network and anisotropic convolution to reduce resource consumption.
> Propose Cross supervision framework and unlabeled image filtering to obtain reliable unlabeled images for semi-supervised training.
> Ablation study for the number of unlabeled data: this experiment shows that UIF effectively selects high-quality unlabeled images.
>
> Response5.1: We thank the reviewer for the important comments. Please see the detailed responses below.
>
> Comment5.2: Add description and figure for network architecture.
>
> Response5.2: Thanks for the comments. We have divided Figure 2 into two figures in the revised manuscript. Moreover, we have added more explanations for Figure2, Figure 3 and Figure 4 in the revised manuscript.
>     As for the details of the network architecture, we adopted the efficientSegNet, and use its strategy to reduce the resource consumption.
>
> Comment5.3: The mDSC of baseline(supervised training) is about 80% or more in this competition, while it's 70.9% in this work. It is recommended to improve the performance of the baseline, and the result will be more credible.
>
> Response5.3: Thanks for the comments. We agree that the baseline performance could be better. In our paper, we resampled all the data into a fixed spacing, i.e., 1×1×3mm. We use cropping strategy to focus on the abdominal region. Moreover, we use the efficientSegNet as our basic segmentation network, which resize the image into a fixed size, and input the whole image to the network for training. Based on this, we adapted dice loss and focal loss as the supervised loss function. Moreover, the proposed solution used the labeled and unlabeled images for training online, and it has the flexibility to switch between semi-supervised and supervised methods. These strategies above may result in a low score at baseline.

---

### Official Review · Reviewer_keeR · 2022-09-20
**Cross supervision using siamese network**

**Rating:** 8
**Confidence:** 4

**Review:**

* Why the best results are not shown in the abstract.
* According to the template, there is no section of segmentation efficiency results and limitation and future work.

---

> ### Author Response · Authors · 2022-10-20
> **We have revised manuscript and provide a point-by-point response to each of the issues raised by reviewer keeR (Reviwer #6)**
>
> Comment6.1: Why the best results are not shown in the abstract.
>
> Response6.1: Thanks for the comments. We submitted SemiSeg-CSSN (50 labeled and 2000 unlabeled iamges) as the final solution for testing, so it is best to present the results of the submitted solution for a fair comparison with other solutions.
>     In the revised manuscript, we have replaced the results in the abstract with the results from the test set, which was provided by the organizers of the challenge.
>
>
> Comment6.2: According to the template, there is no section of segmentation efficiency results and limitation and future work.
>
> Response6.2: Thanks for the comments. We have added the section of segmentation efficiency results and limitation and future work in the revised manuscript.

---

### Meta-Review · Program_Chairs · 2022-09-28

**Recommendation:** Minor Revision
**Confidence:** 5

**Metareview:**

Nice paper. Please address the reviewers' comments in the revised manuscript.

---

> ### Author Response · Authors · 2022-10-20
> **Our responses are prepared based on a point-by-point response to each of the issues raised by the reviewers.**
>
> We would like to thank you for your very constructive and thoughtful comments and useful suggestions, which have greatly improved our manuscript. We have carefully studied each of the comments and revised the manuscript by considering all the suggestions/comments made by the reviewers. Our responses are prepared based on a point-by-point response to each of the issues raised by the reviewers. Major revisions we made include:
> 1.	We rephrased some concepts, words and sentences as the reviewers pointed out .
> 2.	To make the presentation of the paper more completed , we reorganized some sections according to the paper checklist.
> 3.	We added testing results in the revised manuscript.